# Is Responsive Feeding Difficult? A Case Study in Teso South Sub-County, Kenya

**DOI:** 10.3390/nu14214677

**Published:** 2022-11-04

**Authors:** Eleonore C. Kretz, Annet Itaru, Maria Gracia Glas, Lydiah Maruti Waswa, Irmgard Jordan

**Affiliations:** 1University of Applied Sciences Münster, Centre of Competence for Humanitarian Relief, 48149 Münster, Germany; 2School of Public Health and Biomedical Science and Technology, Masinde Muliro University of Science and Technology, Kakamega P.O. Box 190-50100, Kenya; 3Center for International Development and Environmental Research (ZEU), Justus Liebig University Giessen, 35390 Gießen, Germany; 4Department of Human Nutrition, Egerton University, Egerton P.O. Box 536-20115, Kenya; 5Food Environment and Consumer Behaviour Lever Africa Hub, Alliance Bioversity International and CIAT, Nairobi P.O. Box 823-00621, Kenya

**Keywords:** responsive feeding, nutrition security, consumer behavior, Trials of Improved Practices (TIPs), Kenya

## Abstract

Responsive infant and young child feeding as a reciprocal relationship between the child and his or her caregiver is recommended by the WHO but has received less attention than dietary diversity or meal frequency up to now. The current study assessed common (non)responsive child feeding practices and factors that facilitate or hinder caregivers to improve feeding practices in rural Teso South Sub-County, Western Kenya. The qualitative study used focus group discussion (*n* = 93) and Trials of Improved Practices (TIPs) (*n* = 48) to identify challenges and opportunities in household food distribution and feeding practices. Overall, the implementation of responsive feeding practices was feasible for the caregivers. Parents reported mainly positive experiences in terms of the child’s feeding behavior and effects on child health. Traditional beliefs, practices, and cultural norms hindered some households to change intrahousehold food distribution. Households who manage to implement responsive feeding even in food insecure regions should be consulted to (a) improve existing nutrition education messages that acknowledge these cultural norms, (b) to include more responsive feeding information in nutrition education material, and (c) to address gender norms to create awareness of the importance of responsive feeding practices and the need for adequate time allocation for infant and young child feeding.

## 1. Introduction

The development of children’s eating behavior and their dietary intake is influenced by parents, other caregivers, and their social interactions through verbal and nonverbal communication during feeding situations [1,2,3,4,5]. Mutual interactions between caregiver and child are referred to under the term “responsive feeding”, which was defined by Harbron et al. (2013) as a reciprocal relationship between the child and his/her caregiver [5]. These interactions include the child’s verbal or nonverbal communication of hunger and satiety, to which the caregiver responds immediately by offering age-appropriate and nutritious food in a supportive way and appropriate feeding environment. Although the provision of age-appropriate and nutritious food is included in this definition, the focus is usually more on the feeding situation, such as the communication between the caregiver and the child, the balance between assistance and encouragement to eat independently, the way the food is served, or the atmosphere of the feeding situation.

Caregiver feeding styles are presumed to shape children’s preferences and food-consumption patterns. A feeding style characterized by controlling the child’s eating without regarding the child’s choices and preferences was, for example, negatively associated with vegetable consumption in African American and Hispanic preschool children [3]. At the same time encouraging children to eat healthy foods, while giving them choices about eating options as well, was positively associated with the consumption of dairy, fruit, and vegetables [3]. Overall, studies indicate that children benefit from a responsive feeding style regarding growth, eating behavior, and nutrient intake, whereas nonresponsive feeding was often associated with feeding problems as well as under- and overnutrition [4,5].

The practice of responsive feeding is part of the list of recommendations for infant and young child feeding published by the World Health Organization [6,7]. However, recommendations on responsive feeding do not receive the same attention as those on the child’s intake, meal frequencies, or dietary diversity. This trend is also evident for developing countries, where there have been more studies assessing the quantity and quality of children’s dietary intake than the way children are fed, the feeding environment, and interactions with caregivers.

In Kenya, 26% of children, 6–59 months old, are stunted [8]. This stunting rate represents a high prevalence according to the thresholds of the WHO and UNICEF and calls out for public health actions [9]. Research approaches in Western Kenya revealed poor complementary feeding practices with low dietary diversity for children aged 6–23 months [10] as well as other feeding problems such as feeding unhealthy snacks, the provision of watery, nutrient-poor porridge, and early weaning [11,12]. However, knowledge of responsive child feeding practices in Western Kenya is limited, but they are likely to contribute to poor dietary intake and malnutrition among children.

The current study was, therefore, conducted to assess common (non)responsive child feeding practices by targeting children 0.5 to 8 years of age in rural communities in Teso South Sub-County, Western Kenya, and investigate factors that facilitate or hinder caregivers to implement improved responsive feeding practices.

## 2. Materials and Methods

The study is part of a larger study called EaTSANE [13,14]. Details have been reported in a substudy on fruit consumption [15]. In short, this study was conducted in Teso South Sub-County, which borders Uganda in Western Kenya. Sample size was purposively selected (*n* = 8 villages). Each location was selected based on their closeness to the agricultural demonstration and training plots established within the project. Further, families were purposively selected within the selected villages from a previously established cohort, implemented by the principal investigators in the same region between 2015–2018 [15,16]. Thus, eligibility criteria were being a smallholder farmer with a child below the age of eight years and living in one of the selected villages. Participation in the study was on a voluntary basis, and informed written consent was obtained from the participants prior to any data collection.

### 2.1. Data Collection Tools

#### 2.1.1. Focus Group Discussions (FGDs)

Focus group discussions were conducted in four out of the eight project villages and were held at a participant’s home or a public building. The focus group discussions aimed at the determination of common child feeding practices within the community, focusing on possible inadequate practices. Different discussion groups of men, women, and adolescents were purposively formed. The discussion sessions were conducted, with the help of an interview guide, under the guidance of a facilitator, who led the discussions and at the same time ensured a pleasant and open atmosphere [17]. The conversations were held in the local language and the notetaker was responsible for writing down the statements of the participants in English. The discussions were recorded and listened to after the sessions by the facilitators, to add missing information to the notes. The audio recordings were, therefore, not verbatim but summarizing transcribed that were reviewed for quality assurance.

A second round of focus group discussions was conducted at the end of the data collection during evaluation workshops. The focus was on the participants’ experiences and perceptions during and after trials of improved practices (TIPs).

#### 2.1.2. Trials of Improved Practices (TIPs)

The Trials of Improved Practices (TIPs) included 53 households in eight villages for data collection and is an established formative research technique in the field of social and behavior change [18].

The trials were conducted between August and October 2019, within the EaTSANE project. The focus was on how best to improve dietary intake in families with children aged 0.5 to 8 years. Results from how best to improve fruit intake have been reported elsewhere [15]. This study focused on testing the feasibility of recommendations in regard to responsive child feeding practices among primary caregivers. They were offered a choice of recommendations, agreed with the enumerators which recommendation to try out, and were asked to provide details about the choices they made. Stories of successes and failures in implementation were collected and discussed in the recurrent counseling sessions. This process enables insights into barriers and motivators for behavior change to improve dietary practices [15,19].

Facilitators were trained on how to conduct TIPs and counseling of child feeding practices recommendations. In addition, they were trained on how to deal with possible difficult scenarios during the counseling visits as well as the standardized data collection tools (e.g., monitoring sheets). Dialogues between mothers and facilitators (*n* = 4) were conducted in Swahili and other local languages. Refresher trainings were held to discuss (a) challenges in implementation of the recommendations and (b) how to interact with the participants in order to negotiate about the different choices presented during the trials to improve child feeding practices [15].

The TIPs included one baseline and two follow-up visits. As described by Kretz et al. [15], the facilitators assessed the child feeding practices within the participating households (first visit) and recommended practices that needed improvement. Not all caregivers, thus, received the same recommendations. During the next two consequent household visits, counseling continued in order to identify merging issues when it came to putting recommendations into practices. Respective information was documented and translated for data analysis. The interview guides provided to the facilitators included questions about motivators and barriers to change feeding practices and, in particular, the participants’ experiences with the recommendations and their intentions to continue implementing the recommendations in the future.

### 2.2. Child Feeding Practices Tested during the Trials of Improved Practices

Four topics were addressed during the TIPs, based on inadequate practices determined during the observations made in the previous focus group discussions: (1) nutritious meals and snacks, (2) fruit consumption and availability, (3) porridge quality, and (4) feeding practices and priorities during mealtimes. In total, the catalog included 13 recommendations for child feeding practices, of which 3 recommendations particularly addressed responsive feeding practices [15]:“Make child feeding a priority in your household. Serve young children first. Make sure they get and eat their share” [15].“Separate the child’s bowl from the mother’s in order to know how much the child has eaten” [15].“Interact with the child during mealtimes and actively and lovingly encourage the child to eat; do not force or threaten your child to eat” [15].

### 2.3. Data Analysis

The data collected during the focus group discussions and the TIPs were analyzed by applying a structuring qualitative content analysis [20]. Like in the larger EaTSANE study, the data were primarily categorized following the thematic structure of the interview guides. Subcategories were inductively and gradually derived from the respondents’ answers. The categories were defined, and a coding guideline developed. The coding process was done with support of the open-source software QDA Minor Lite v1.4.1, by Provalis Research. Following the approach in the larger study, coding was validated with the support of a second coder. During the second coding round, disagreements were discussed amongst the coders. Definitions for specific codes were discussed, and the coding guideline was revised accordingly [15].

Finally, Cohen’s Kappa confirmed acceptable intercoder agreement with a coefficient of 0.82 ± 0.02 (SE); (*p* ≤ 0.001) for data analysis of the focus group discussions, 0.78 ± 0.04 (SE); (*p* ≤ 0.001) for data analysis of the TIPs data, and 0.82 ± 0.03 (SE); (*p* ≤ 0.001) for data analysis of the workshops, respectively. The calculations were done with IBM SPSS Statistics Version 27.

In addition to identifying the main barriers and motivators for implementing the tested practices, a descriptive analysis showed how many caregivers used inappropriate practices and how many successfully tested the recommendations.

## 3. Results

### 3.1. Main Characteristics of Study Participants

Each of the 12 focus group discussions consisted of 5 to 11 participants. The mean duration was about two hours. In total, 93 participants took part in the discussions, with 29 participants taking part in the women’s groups, 26 in the men’s groups, and 38 in the youths’ groups.

The TIP respondents were women aged between 22 and 58 years. They were primary caregivers of up to four children aged 0.5 to eight years.

Out of the 53 households that participated in the study, 48 households were included in the final data analysis (Figure 1). In total, there were five dropouts during the household visits. Two households moved away and were, thus, not eligible to participate anymore, and two households withdrew their approval to participate due to a lack of interest. One household was excluded from the data analysis. A change in the family situation required extra counseling by the facilitators, and the team decided that this may have influenced the data.

### 3.2. Common Child Feeding Practices

The focus group discussion revealed that mothers have challenges in four core areas of responsive feeding: (1) serving priorities during mealtimes, (2) who is involved in feeding the children, (3) sharing a plate, and (4) strategies within the family to improve the children’s food intake.

#### 3.2.1. Serving Priorities within Households during Mealtimes Hinder or Support Adequate Food Intake

The discussion showed a high occurrence of the practice to serve the father or husband first and the youngest child last:
“*I am served first, my wife serves herself then eldest son to youngest child*”.(FGD Men)
“*Father [is] served first, then eldest son, then mother and other children come last*”.(FGD Youth)

However, this was not always the case. In some households, children were served before the parents (FGD Men), and statements showed that the youngest child was given priority:
“*Youngest child first, older children, my husband and myself*”.(FGD Women)

One reason for the preference of serving young children first was attributed to the child disturbing the others while eating:
“*I serve the youngest child first because she won’t allow other people to eat as she waits, then I serve my husband, other children, then I serve last*”.(FGD Women)

Another social aspect of the serving practice is a potential reunion as a family for mealtimes. Answers indicate that family members tended to eat separately:
“*I serve my husband and take his food to the main house then I serve myself and children come later*”.(FGD Women)

However, others reported a communal meal:
“*I get food on the table so we just eat together as a family*”.(FGD Men)

#### 3.2.2. Different Persons Involved in Feeding May Enhance Child’s Food Intake

Children were not only fed by their mothers but also, particularly in the mother’s absence, by other family members or household members. Older siblings seemed to play an important role as caregivers regarding feeding. Nevertheless, men were mentioned to be involved in feeding too:
“*There are children who eat well only when they are fed by their fathers*”.(FGD Men)

The role of grandmothers and grandfathers was not discussed.

#### 3.2.3. Food Served on a Shared Plate Hinders Adequate Food Intake

The practice of sharing a plate with young children is important, as it is associated with the advice to observe a child’s food intake. The discussions showed that families applied different practices when feeding children above and below the age of two years. The practice of sharing a plate with children over the age of two years was mentioned noticeably less frequently. Participants stated that children under the age of two years were fed from a collective plate, mostly shared with the mother and sometimes with the father or other children. A few reasons were named for this practice for both age groups, above and below two years of age (Table 1). During the trials of improved practices, most mothers did not report about shared-plate eating. In four households, the practice was carried out. In one case, twins of 17 months of age shared a plate during mealtimes; in the other cases, the mothers shared a plate with their child.

Linked with a separate plate for children was the assumption that they were already able to pick food for themselves or eat independently without the need for assistance. However, it was stated that the need for help in feeding was not an exclusion criterion to being served on a separate plate. Other participants emphasized that children in general should be fed from their own plates. Common factors cited for both practices were the child’s age and education. Assistance was granted to children independent from a shared or a separate plate, but it was considered to be time-saving when the child ate by him-/herself from its own plate.

Learning social skills was associated with eating on a shared plate, whereas others viewed the pressure placed on children to receive a fair and adequate share of the meal as more negative.

#### 3.2.4. Caregiver Strategies for Child’s Food Intake Include Positive and Harmful Practices

There were five main categories of answers about ways to ensure adequate food intake when a child refused to eat. They ranged from positive encouragement such as providing company, showing care and affection, or searching for alternatives to violent behavior. Force, beating, and threatening the child were mentioned by all different groups (women, men, and adolescents) and seemed to be culturally acceptable. However, these statements were mostly made with laughter, indicating some insecurity. On the other hand, encouraging behavior was also found in every group, showing that caregivers knew and practiced other feeding styles. Solutions to boost a child’s appetite also included harmful practices such as offering the child glucose or multivitamins prior to the meal:
“*I give him glucose to boost his appetite first*”.(FGD Women p7: l.299)

Table 2 provides an overview about the strategies to improve food intake among children, an explanation, and an exemplary quote for each strategy.

### 3.3. Implementation Rate of Recommendations

The recommendations were developed based on the focus group discussions’ findings and provided among those households that showed inadequate practices during the first visit. Table 3 provides an overview about which inadequate practice was addressed and which recommendation was given to how many households. All households that received recommendations regarding responsive feeding and priorities at mealtimes implemented the practices for the duration of the trials. Modifications to the recommended practices were made in three households. Instead of changing the serving order during meals to start with the children, the mothers ensured that young children received and ate an appropriate portion of the meal.

The TIPs identified several factors that facilitate and hinder the implementation of the recommended practices, including the possible influence and reactions of family members, especially the children themselves, to these recommendations. The children’s preferences were considered important in the making of food choices. The caregivers reported that their children preferred certain foods, e.g., tea instead of porridge, soup instead of solid foods, or just biscuits. To avoid conflict, the mothers followed the children’s preferences and did not offer healthy food as an alternative [15]. Nonetheless, a child’s preferences did not always lead to an adaption of food choices, and some caregivers resorted to the use of force. One mother explained her violent behavior toward her child with the child’s refusal to eat on the basis that “*the child does not like most of the foods offered*” (woman, 27 yr (with child, 4 yr)). Other caregivers also mentioned the use of force when the child refused to eat. The practice of the mother sharing a bowl with the child was explained with the child’s inability to eat alone:
“*[The] child [is] still small and cannot feed on its own*”.(woman, 32 yr (with child, 1 yr))

Difficult experiences that were mentioned after trying the improved practices concerned the reactions of the children. One mother reported that her child started to play with the food when served on a separate plate (woman, 32 yr (with child, 1 yr)). Other difficulties were faced when the children refused to accept the improved practices, mostly due to foods that they disliked:
“*The child completely refused to take porridge*”.(woman, 45 yr (with child, 7 yr))

However, the reaction of the children to the new practices was often positive. Mothers reported that children took the enriched porridge without any difficulty and had no problem swallowing one with a thicker consistency. Children were able to eat from separate plates and started to pick food on their own.

The reactions of other family members were also mentioned as positive experiences during the trial. When mothers tried to change the serving order of meals, the men approved of being served after the children.

Lifestyle, daily routines, and habits hindered the change of current inadequate practices. It concerned, e.g., the serving order in the household:
“*Serving starts from the oldest to the youngest*”.(woman, 27 yr (with child, 4 yr))

Furthermore, the practice of serving the fathers first was explained by the traditional role allocation between men and women. Many mothers stated that they served them first because the “*husband is the household head*” (woman, 37 yr (with child, 2 yr)). Convenience and time constraints also hindered serving the child food on an individual plate:
“*It is too much work having two plates at once since I eat with my son at the same time*”.(woman, 26 yr (with child, 2 yr))

The ability to take extra time for child feeding facilitated the implementation of the improved practices. Regarding encouraging a child through interactive mealtimes and matching food choices to the child’s preferences better, one mother explained that she was “*willing to try because she has time*” (woman, 27 yr (with child, 4 yr)). This was not always the case, as caregivers often reported time constraints and times of being absent from home, which hindered some caregivers’ ability to test the improved practices. Having mothers in charge of meals helped them implement new behaviors, such as giving preference to infants at mealtimes:
“*[I am] willing to try that since I am the one who serves the meals*”.(woman, 27 yr (with child, 4 yr))

The knowledge gained through the study’s nutrition counseling about the nutritional and health benefits for their children acted as a motivating factor to try the improved practices. Respondents were
“*willing to do so because [they] now understand[] the importance of child feeding*”.(woman, 27 yr (with child, 4 yr))

Perceptions about children’s progress in learning to eat independently, as well as the opinion that children should enjoy eating, also moved caregivers to action. Other perceptions functioned as barriers to change a common practice. These include perceived respect for the husband, related to serving him first:
“*[I] always wants to serve my husband first because of respect*”.(woman, 32 yr (with child, 1 yr))

Other respondents did not find it necessary to serve children first as
“*even though she serves the husband first, the child still gets adequate share of the meal*”.(woman, 32 yr (with child, 5 yr))

Caregivers also expressed their emotions about the way they feed their children. One mother stated about her forceful behavior toward the child that
“*[I] sometimes also feels bad about doing it and also beating had negative effects on child feeding*”.(woman, 52 yr (with child, 7 yr))

Regarding priorities during mealtimes, a respondent acknowledged:
“*I don’t like when the child cries after seeing my husband eating food before her*”.(woman, 30 yr (with child, 4 yr))

As the households experimented with improved feeding practices, they reported experiencing and observing different positive outcomes regarding their children’s food intake, health, and behavior. The children were said to have improved their eating behavior and to have been more satisfied, resulting in fewer disruptions for parents:
“*When the children are served first, they don’t have to disturb their father*”.(woman, 37 yr (with child, 2 yr))

Caregivers also reported about encouraging interactions they had with their children during feeding situations and about the children’s learning progress, to eat more independently while being supervised more closely. Even weight gain was reported:
“*Since the child started eating from his own bowl, he has really improved and added some kilos*”.(woman, 26 yr (with child, 2 yr))

### 3.4. Households’ Reflection of Experiences Made during TIPs

Caregiver responses, during the evaluation workshops six weeks after the last TIPs household visits, indicated that all responsive feeding practices were remembered and partially implemented, except for prioritizing child feeding within the household. The men observed that fewer children were forced to eat but more were assisted to eat. However, the men did not report noticing whether the “feed the child from his/her own plate” recommendation had been implemented. The reported experiences with the implementation of improved responsive feeding practices were consistently positive. Both women and men had also observed positive health effects on the child and changes in the child’s behavior such as a greater satisfaction and less crying (Table 4).

## 4. Discussion

In general, there is limited information in the literature on responsive feeding practices compared to other feeding practices such as dietary intake or diversity, especially in developing countries. This study showed how inadequate responsive feeding practices, including shared-plate eating, forced feeding, and the low priority of child feeding within households may be successfully addressed.

The practice of children sharing their plate when eating or feeding was evaluated differently by the study participants. Shared-plate eating was reported as a common practice in other low-income countries too [21]. Burrows et al. revealed variations within this practice, showing that food is either consumed directly from a central dish or served from a main plate to additional plates that are shared by two or more people [21]. In the current study, respondents mainly referred to children sharing a plate with the mother or the father and shared plates among siblings. The argument that one can monitor the amount of food the child has consumed was not mentioned, although it presents a major challenge, not only for the caregiver to estimate whether the child consumed an age-appropriate amount of food but also for the general assessment of dietary intake within research approaches [21].

The complex distribution of food within households has been widely described in the literature under various aspects. For example, a male household head in Guatemala was granted a bigger share of the protein components than the rest of the family members, whereas the mother received a greater amount of the calories [22]. Concerns about inequalities in food distributions were expressed within the current study, with some participants stating about their child that “*if served on the same plate with others they will finish for him*” (FGD Women). In contrast to the negative effects of shared-plate eating, with smaller children getting smaller portion sizes, a case-control study in Nepal suggested that the practice is a rather beneficial eating behavior [23]. The Nepal study indicated that shared plates are associated with the consumption of a greater variety of foods and larger portion sizes compared to individual-plate feeding in general, whereas there were no differences in the consumption of carotenoid-rich fruit, vegetables, or meat. However, the authors also stated that shared-plate eating mainly occurred during main meals and only sporadically during snacks, so, subsequently, shared-plate meals were more likely to include a greater variety of food groups [23]. Another interview-based study in southeastern Nigeria described sharing plates among siblings as a parental strategy to encourage competitive eating among their children and, subsequently, to eradicate picky eating [24].

Another aspect of shared-plate eating concerned assisting the child to eat or allowing the child to gain experiences while eating on her/his own under supervision. In this study, assistance was independent of whether the food was served on a shared or separate plate but was linked to the caregiver’s attitude toward care and supervision during mealtimes. The argument for a separate plate, “*because he eats while playing thus taking too much time*” (FGD Women), indicates a lack of understanding about providing interactive and playful mealtimes and resulted, most likely due to time constraints, in limited care and assistance for the child. At the same time, the child loses their autonomy and may fail to appreciate satiety signals [25].

### 4.1. Persons Involved in Feeding

As a rule, mothers were the main caregivers responsible for child feeding in the households. In the mothers’ absence, other family members were in charge. The same groups of persons in different variations were described as caregivers in regard to feeding, in various studies [26,27,28]. A study by Wawire (2017) described the same caregivers for children in Migori and Kisumu, Western Kenya, with the additional explanation that mothers mostly did not leave their children without having a meal prepared beforehand [11]. The responses given in the focus group discussions were related to persons taking over the mothers’ task in their absence, though it was not clear to what extent these persons assisted in general. During the discussion about caregivers’ strategies for the children’s food intake, it was mentioned that other family members took over to feed the children because this resulted in better eating behavior from the children. It appears that interaction with certain individuals during feeding could increase food intake, highlighting the relationship between a specific feeding person and the child in terms of responsive feeding.

### 4.2. Serving Priorities during Mealtimes

The serving order of meals within households seemed to be determined by traditional role allocations. The tradition to have a specific serving order, which privileged the male head of the household to be served first, has also been described for Guatemalan families, where it was associated with inequal intrahousehold food distributions [22]. In contrast, an observational study in a rural area in mid-Western Nepal reported about an intrahousehold serving order in which small children, regardless of their sex, had absolute priority [29]. The children were traditionally served before the male head of the household, and adult women served themselves last [29]. Although the serving order is not necessarily associated with an inadequate share of food for different household members, a preference given to children is seen as an indicator of adequate childcare and as an important contribution for the children’s nutritional status [30]. A cross-sectional study in rural Nigeria showed that children were less likely to be stunted when they were given the priority of receiving a more diverse diet, compared to children in households where male or female adults were preferred in terms of food variety [30]. The focus group discussions in terms of serving orders and priorities were inhomogeneous, leading to the assumption that community members differ in their traditional practices and values or may have been exposed to similar nutrition education messages in this regard already. The families who already prioritized child feeding within the intrahousehold food distribution or were willing to change the serving order during the TIPs could become peers within parent–child-care groups, which would support a sustainable behavior change.

### 4.3. Caregivers’ Strategies for Children’s Food Intake

Both responsive and nonresponsive feeding styles were a topic in the focus group discussions as well as in the TIPs. The responsive feeding style is characterized by a positive interaction with the child during feeding/eating, which can include encouragement in any form, conversations, playing games, smiling, and eye-to-eye contact, among others, bearing in mind the interest of the child [5,31]. Encouragement through company, showing love and affection, or promised rewards was mentioned by the study participants. Likewise, the search for alternatives showed the caregiver’s concern about the interest of the child, by providing extra food. Moreover, feeding more frequently in small amounts or letting other people, who seem to have a positive influence on the child’s eating behavior, feed the child can be considered a way out of a problem that has started at an earlier parenting age [31]. It is questionable whether the strategy of waiting until the child is hungry can be considered responsive. On the one hand, it fulfills the criterion of being oriented toward the expressed needs of the child; on the other hand, the underlying motivation for this could be a lack of willingness to take the time to sit down with the child for an interactive meal.

Habron et al. (2013) classified nonresponsive feeding styles into three types: the indulgence type lets the child control the situation, the uninvolved type ignores the child during mealtimes, and the restricting type excessively controls and dominates feeding times [5]. The latter can be recognized in the study population by the reported behavior of forcing the child to eat. Porridge was especially named as being forced on the child. Pelto et al. (2003) reported this practice as well in one of their case studies, in which mothers in Nigeria fed the child porridge by hand. They held the child’s nose when they refused, which forced them to swallow [26]. These practices are not easily reported and are least often addressed in nutrition education approaches, as they remain a culturally sensitive aspect of child feeding. During the TIPs, most mothers reported that their children ate without difficulties, but, upon further probing, they reported using strategies to increase the child’s food intake, such as force. These practices were also described for complementary feeding practices in Migori and Kisumu, Western Kenya, where mothers used force in the case of the child’s refusal but did not feel comfortable doing so [11]. Violence and nonresponsive behavior can lead to feeding situations characterized by tedious interactions and a lack of trust in the relationship between caregiver and child [5]. Furthermore, pressuring practices in child feeding were predominantly found to be associated with higher rates of picky-eating behavior, a lower interest in food, and a lower intake of food [4].

### 4.4. Reactions and Influence of Children

In the current study, the children’s preferences or dislikes for certain foods influenced the food choices made by their mothers. Children’s reactions to the improved practices facilitated or hindered their implementation. This was to be expected, since the children were the ones affected by a change in feeding style. Still, the reactions from the children in the current study to new practices were predominantly positive and approving and facilitated the implementation. Unproblematic adaptions to new practices were also reported for children in studies using TIPs in Rwanda and Uganda [32,33].

Some children in the current study were reported to refuse certain foods or to be slow or picky eaters. Whether this was a result of nonresponsive feeding practices at an earlier age could not be assessed in this study. However, in these cases, caregivers should be encouraged to introduce new food items several times to allow a child to become familiar with the food [4,5]. Foods should also be presented early in life to achieve higher acceptance, leading to higher frequency of consumption of these foods later [4,34]. This is especially important when nutrition education aims to improve dietary diversity, which often involves introducing new recipes and flavors. Within the study population, not even half of the children aged 6–23 months met the minimum dietary diversity score for children, but an age-related increase in the integration of more food groups into children’s diets was noted by Waswa et al. (2015), which was confirmed within the baseline survey of this study [10]. Besides the introduction of a variety of foods in early life, a child should be given the opportunity to explore and learn to eat, while being assisted by their mother. Likewise, a child playing with food should not be considered as a hindering factor to offering a meal on a separate plate. This was explained during the TIPs with the concept of an “interactive mealtime”, where the feeding situation is supposed to be relaxed and playful, while mother and child have eye-contact during their feeding interaction [5].

A cross-sectional study in Vietnam investigated caregiver feeding styles with children 12 or 18 months of age, using video recording during mealtimes. Positive verbalization from the caregiver was associated with a higher acceptance of food, whereas threatening verbalization and force led to higher rates of rejection of food [35]. Similar experiences were reported by caregivers in the current study, when a child was positively encouraged to eat:
“*The child eats even better than when force was being used during meals*”.(woman, 27 yr (with child, 4 yr))

### 4.5. Lifestyle and Daily Routines

Time was rarely explicitly mentioned to influence responsive feeding practices but became obvious through statements about the convenience of a practice, the mothers’ absence from home, or the expressed disagreement with the child playing with food during mealtimes. A statement by one of the mothers, that she had time to try out the practice, shows that this does not mean the norm. Women in developing countries often face a double or even a triple burden of work with housework, childcare, and subsistence farming activities. In addition, women are increasingly involved in paid employment [36]. Heavy workloads, therefore, rival with the time required for adequate childcare and feeding practices. “Competing interests for caregivers’ time” is considered as a barrier to implement responsive child feeding, highlighting the need for development to generate extra time to allow for the implementation of improved infant and young - child feeding practices [28,32,37,38].

### 4.6. Perceptions and Tradition

The respondents’ perceptions of the current and improved responsive feeding practices were identified as facilitating and hindering factors. According to social psychological theories, perceptions are attributed with an important influence on behavior. Even if the objective consequences of an action were fairly certain, the intentions for a particular behavior would depend on how the person interprets these consequences [39]. Perceptions of the traditional roles of men and women influenced some mothers regarding the serving order in the family. They refused to serve the child first because they felt it was unnecessary or feared that this would be perceived as a lack of respect toward their husband, as the head of the household. Unlike the participants who changed the serving order with their husbands’ consent, there was no information on the influence or reaction of the husbands of the refusing participants. None of the mothers mentioned difficult or discouraging reactions from their husbands. This could be explained by the relatively low participation of men in child nutrition decision-making in developing countries [27]. In more traditional societies, a tendency toward gender-specific roles was portrayed, with childcare seen as the responsibility of women. Therefore, despite being in most cases the “household head” in the current study, men were rarely involved in the decision-making concerning child feeding. However, although some women referred to themselves as the person in charge of meals, which facilitated the implementation of new feeding practices, others did not view themselves as confident enough to change the traditional serving order. Traditional norms and gender roles were, therefore, an important influencing factor and need to be taken into account in programs like this one, by involving husbands and community elders to gain their understanding and support for the concerns of children and women [27]. Several other studies showed that the sociocultural environment significantly influences child feeding practices [40,41,42]. Food taboos are increasingly and successfully addressed in nutrition education, but the household dynamics of who is served first and last or whether a child is forced to eat, for example, tend not to be addressed. Maybe this is a result of nutrition education up to now being a rather women-focused strategy. This requires more gender-specific nutrition education programs that also include men in the discussion of infant and young child needs, in order to combat malnutrition and improve fathers’ views of infant and child needs.

### 4.7. Positive Outcomes of the Improved Responsive Feeding Practices

The study participants reported several positive effects such as improved child’s food intake, health, or behavior when trying out the improved feeding methods. The perceived benefit to the child’s health was described as the most common facilitator for the caregivers in other studies too [32,33,43]. The thicker consistency of porridge was perceived to contribute to a longer satiety, more energy, and weight gain [33,43]. Similar statements were made in the current study and highlight the underlying motivational factors for the mothers to guarantee the health and well-being of their children. These positive observations of the caregivers were also reported in the focus group discussions during the evaluation workshops. Mothers and fathers independently reported that they had seen changes in their children regarding a better food intake, positive health effects on the child, and a more satisfied behavior. As the period between the first household visits for the trials of improved practices and the workshops was about three months, positive effects could have been possible, but no measurements took place at the end that would have confirmed them (such as weight gain). The given statements reflect positive expectancies about the recommended responsive feeding practices and can function as motivators to implement the practices.

### 4.8. Limitations

This study focused on the implementation of child feeding recommendations in accordance with the WHO and did not include a systematic assessment of responsive feeding practices using instruments such as standardized questionnaires [44]. We also studied a wide age range of children. However, practices such as sharing plates, household priorities in serving meals, or encouragement while eating, as well as the general atmosphere at mealtimes, affect younger and older children in Kenya equally. All results of this study rely on the self-reporting of the respondents and are, thus, exposed to bias. Furthermore, data collection was conducted in the local language, with answers noted down in English in a summarized form, instead of performing a verbatim transcription following a translation. Detailed information could have been lost during the process, and there was a possibility of a subjective interpretation of the respondent’s answers by the facilitators.

The TIPs were not only carried out to collect data on the issue of child feeding practices but also on two other subjects for the research project. This implied counseling on three different topics during one household visit and resulted in a high number of recommended improved practices that the women agreed upon. The long counseling process might have exceeded the absorption capacity for information of the women, and the additional practices might have influenced the implementation of the ones concerning child feeding practices, by competing with the women’s time and other resources.

One has also to consider that this study followed nutrition counseling about dietary diversity, which occurred four months earlier. Information about the effects on the child had already been pointed out by the parents at that time, so respondents may have repeated these results. The effect of a method bias within the research approach could have occurred, if respondents pursued the practices for social approval through desirable behavior. This “social desirability” refers to the tendency to present oneself in a favorable way, despite one’s true perceptions and feelings [45]. In the focus group discussions, there is also the likelihood of respondents agreeing with what another participant said, as well as the possibility of an answer being given only because the participant felt obliged to do so [17].

The deductive formation of categories depending on the interview questions was used for structuring. These questions were researched beforehand through the literature and observations. This is an established method. Still, it is possible that certain feeding practices were not discovered through this method, as we concentrated on practices already mentioned in the literature and what we have seen. The answers of the participants, in contrast, were then coded inductively without any predefined structure to minimize the potential bias.

## 5. Conclusions

Overall, the implementation of responsive feeding practices was feasible for the caregivers. They reported mainly positive experiences in terms of the children’s feeding behavior and effects on the children’s health. Traditional beliefs, practices, and cultural norms posed a great challenge in changing the established feeding practices within some households, although not all.

There is, therefore, a need to involve households that are practicing adapted diets in food-insecure regions, such as those in this study, in the development of nutrition education interventions to improve existing messages. These messages should address cultural norms, include more information about improved diets, and address both genders to create more awareness of the importance of adapted feeding practices and the need for appropriate timing of infant and young child feeding.

## Figures and Tables

**Figure 1 nutrients-14-04677-f001:**
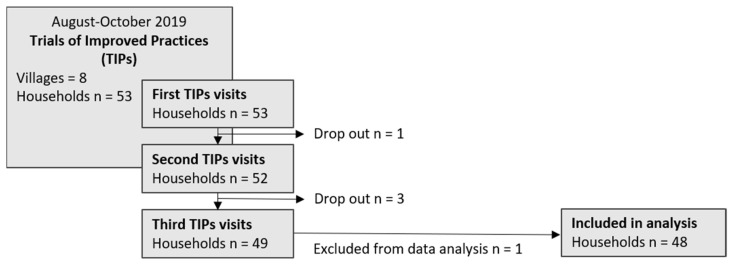
TIPs process and dropouts [15].

**Table 1 nutrients-14-04677-t001:** Reasons for using a shared or a separate plate for children *.

Reason	Explanation	Example	<2 yrs^1^	>2 yrs^2^
**Reasons for a shared plate**				
Tradition	It was stated that it was a common way to eat, as it had always been practiced.	“*[Children] never had own plates*” (FGD Men p4: l.164 col.1); “*[They share the] same plate with me since they are my children*” (FGD Men p4: l.183–184 col.2).	√	√
Age	Children were perceived to be too young to eat from a separate plate.	“*Because they are young*” (FGD Men p4: l.162 col.1)	√	
Eating well	Better food intake by the child was attributed to the practice.	“*When children eat with their mother from the same plate they will eat well*” (FGD Women p5: l.199–201 col.1)	√	
Assistance	The practice was stated to enable the mother to help her child eat, which she could easily do this way.	“*I eat with him from the same plate so that I can help him feed*” (FGD Women p5: l.202–203 col.1)	√	
Training	Children were assumed to be trained by the practice in terms of social behavior.	“*Eats with other people from the same plate so that he/she doesn’t become mean*” (FGD Women p5: l.195–197 col.2)		√
**Reasons for a separate plate**				
Age	Children were perceived to be too young to share a plate or old enough to eat from a separate one.	“*We feel she is still too young to eat from our plates.*” (FGD Men p4: l.171–173 col.1)	√	√
Ability	Children were perceived to be able to eat by themselves.	“*Because they can eat by themselves*” (FGD Men p5: l.188 col.2); “*If they can pick by themselves, they are served on their own plate*” (FGD Men p4: l.181–182 col.1).	√	√
Less pressure	The inability to eat on their own in a timely manner was assumed to put children under pressure when a plate was shared with others, e.g., older children.	“*Because if served on the same plate with others they will finish for him*” (FGD Women p5: l.219–221 col.1); “*Because if he/she shares with other people he eats very fast*” (FGD Youth p5: l.237–238 col.2).	√	√
Cooling the food	It was stated that the food should cool on a separate plate before being served to the child.	“*So that the food cools down a little bit*” (FGD Women p5: l.208–209 col.2); “*It shouldn’t burn [the child]*” (FGD Women p5: l.205–206 col.1).	√	√
Preference of the child	Children were assumed to prefer their own plate or to be used to it.	“*He cries if you don’t serve him food from his own plate*” (FGD Men p4: l.178–179 col.1)	√	√
Assistance	Children were perceived to need assistance, which was provided.	“*Because he can’t share with others and can’t feed by himself*” (FGD Women p5: l.207–209 col.1)	√	
Vulnerability	It was assumed that young children were too vulnerable to share a plate because of a susceptibility to infections.	“*Because they are very sensitive and so they should be served from their own plates*” (FGD Men p5: l.191–193 col.1)	√	
Portion size	Children were assumed to need to be served only a small portion size.	“*Because he needs to eat little*” (FGD Women p5: l.204–205 col.1)	√	
Time saving	Allowing the child to eat independently from his/her own plate was experienced as a time saver for the caregiver.	“*Because he eats while playing thus taking too much time*” (FGD Women p5: l.216–217 col.2)		√
Training	Children were assumed to be trained how to eat independently.	“*As to make them used to their food in their plates*” (FGD Men p4: l.163–164 col.2)		√
Hygiene	Sharing a plate was seen as a hygienic problem when children partook in a communal meal.	“*Sometimes they have a running nose and they pick their nose as they eat and even play in the midst of eating*” (FGD Men p4: l.172–175 col.2)		√

* relevant age group is indicated by “√”; yrs = years; ^1^ valid for children < 2 years of age, ^2^ valid for children > 2 years of age. *Please note: the different reasons have been grouped into categories and highlighted in bold respectively.*

**Table 2 nutrients-14-04677-t002:** Caregivers’ strategies to improve children’s food intake.

Strategy	Explanation	Example
**1. Encouragement**		
Provide company	Company was provided by family members or friends who sat with the child and motivated her/ him to eat.	“*I call her friends and since they eat, she also eats*” (FGD Men p6: l.270)
Show care and affection	A positive effect on the child’s eating behavior was experienced by showing love and affection to the child.	“*I wash the baby, soothe him a little bit then he will eat*” (FGD Youth p7: l.331–332)
Reward the child	The child was rewarded for a good eating behavior.	“*I show her what she likes and promise to give it to her only if she eats*” (FGD Youth p7: l.334–335)
**2. Search for alternatives**		
Provide other foods	Caregivers provided extra foods that the child was more likely to eat or mixed disliked foods with the child’s favorite ones.	“*I provide other alternatives that they like*” (FGD Men p6: l.273)
Feed frequently small meals	The child was frequently fed small meals.	“*I give small frequent meals*” (FGD Men p6: l.283)
Let other people feed	Instead of the mother herself, another person took over the feeding.	“*I let other people feed him because he doesn’t eat well whenever I am the one feeding him*” (FGD Women p7: l.317–318)
Feed only when hungry	Caregivers waited to feed the child until she/he was hungry and subsequently ate voluntarily.	“*I let her play until becomes hungry and then she will eat*” (FGD Women p7: l.306)
Boost the appetite	Caregivers tried to boost the child’s appetite by offering sugar or multivitamins.	“*I give him glucose to boost his appetite first*” (FGD Women p7: l.299)
**3. Violent behavior**		
Force	Caregivers forced the child to eat, in particular when porridge was fed to the child.	“*I force them to eat porridge from my palm*” (FGD Women p7: l.300)
Beating	Caregivers reported beating the child if they refused to eat.	“*I beat them up until they take the porridge or eat the food*” (FGD Women p7: l.301)
Threatening	Caregivers threatened the child with a punishment that included beating or withholding favorite things.	“*I scare him/her with a cane*” (FGD Men p6: l.276)
**4. Lures**		
False promises	False promises were mentioned as a way to lure the child to eat well.	“*I lie to her that if she eats her food, I will buy her mandazi*” (FGD Women p7: l.324) (note: mandazi is an African donut)
**5. Others**		
Observe and try to solve the problem	Caregivers observed the child to find out about underlying problems that could be addressed.	“*I find out why my child doesn’t want to eat and then address the problem*, *e.g., when sick, I take him for medicine*” (FGD Men p7: l.284–285)

Please note: the different strategies have been grouped into categories and highlighted in bold respectively.

**Table 3 nutrients-14-04677-t003:** Key recommendations and implementation rates of inadequate practices addressed in the TIPs.

Inadequate Practice	HH with Inadequate Practice	Recommended Improved Practice [15]	HH Implementing the Improved Practice	HH Modifying the Improved Practice	HH Planning to Continue with the Improved Practice
	*n*		*n* (%)	*n* (%)	*n* (%)
Children are not given priority when food is served; husband is served first.	10	Make child feeding a priority in your household. Serve young children first. Make sure they get and eat their share.	10 (100%)	3 (30%)	5 (50%)
The child is fed from the mother’s plate; children share one plate.	4	Separate the child’s bowl from the mother’s in order to know how much the child has eaten.	4 (100%)	0 (0%)	3 (75%)
The child is forced to eat; the child is left alone to eat.	4	Interact with the child during mealtimes and actively and lovingly encourage her/him to eat; do not force or threaten your child to eat.	4 (100%)	0 (0%)	2 (50%)

**Table 4 nutrients-14-04677-t004:** Experiences and perceptions regarding the changes made in responsive feeding practices, since the TIPs household visits mentioned in the evaluation workshop *.

Experiences and Perceptions Regarding the Changes Made in Child Feeding Practices	Example	Mentioned by Women	Mentioned by Men
**Child’s food intake**			
Child is eating on her/his own without force.	“*My child is eating at his own without being forced*” (W_FGD Women1 p2: l.71–72 col.2)	√	
Mother is able to monitor the child’s food intake.	“*I did not know how much my child could eat but right now am sure of how much he is eating*” (W_FGD Women1 p4: l.143–146 col.2)	√	
**Health effects on the child**			
Child is healthier.	“*The bigger children were finishing food for the smaller one and his health was not good but now he feeds slowly at his own pace and the body is ok*” (W_FGD Women2 p5: l.217–222 col.2)	√	√
**Child’s behavior**			
Child is satisfied.	“*I used to force my child to eat but nowadays I negotiate with my child and even help him feed. The child used to cry even sleeps without eating but nowadays he eats very well knowing the next day he will get a good meal*” (W_FGD Women2 p5: l.205–208 col.1 and 2)	√	√

* responses that were given by women or men are indicated by “√”. Please note: the different experiences and perceptions have been grouped into categories and highlighted in bold respectively.

## Data Availability

The raw data supporting the conclusions of this article will be made available by the authors, without undue reservation. The data are not publicly available due to the privacy restrictions of the farm families.

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
