# Peer review of "Is Responsive Feeding Difficult? A Case Study in Teso South Sub-County, Kenya"

_nutrients, 2022, doi:10.3390/nu14214677_

Round 1

Reviewer 1 Report

This is a well-written manuscript and the methods are well-described. However, the age group of children the study aimed to target is unclear. Feeding practices for children will be different based on age groups. Please clearly mention the age group in the introduction and methods.

The methods of analysis state, ‘data were primarily categorized following the thematic structure of the interview guides’, line 149-150, P4. Could this have brought any bias to the coding and themes formation prior to the analysis? If so, what methods were adapted to address the bias.

Discussion in its present form is long and reads like an extension of findings. The sub-headings in the Discussion are helpful and can be possibly used in the results/findings section.

Lastly, culture plays a major role in feeding and childcare practices in countries with collectivist cultures such as Kenya. While this is mentioned in the Discussion section under ‘Perceptions and tradition’, some more engagement with how culture influences feeding practices and its outcomes in the Discussion section might be useful. 

Author Response

Dear Reviewer,

Thank you for your positive response and your valuable input on this manuscript, which has helped us to improve it. We have revised the manuscript considering your recommendations. As this study is part of a larger study, we were asked to revise the language in some sections to reduce similarity (method section) to a publication we wrote and cited already on a different topic but from the same project. We have revised the text and still refer to the relevant publication without reducing the information needed to understand the study. We hope that the revisions meet with your approval.

Reviewer: This is a well-written manuscript and the methods are well-described. However, the age group of children the study aimed to target is unclear. Feeding practices for children will be different based on age groups. Please clearly mention the age group in the introduction and methods.

Authors response: We have included some more information about the age group. Feedings practices are indeed different based on age groups. However, in this study we included all children up to eight years of age. Recommendations for responsive feeding practices were emphasized for the age group 6-23 months, but did not exclude older children. The practices of shared plate eating, household priorities in serving meals or encouragement and the atmosphere while eating concern younger and older children in the Kenyan environment equally.

We have included the information of the age range to introduction and method and discussed the difficulties involved with the large age range in limitations.

Reviewer: The methods of analysis state, ‘data were primarily categorized following the thematic structure of the interview guides’, line 149-150, P4. Could this have brought any bias to the coding and themes formation prior to the analysis? If so, what methods were adapted to address the bias.

Authors response: The deductive formation of categories depending on the interview questions was used for structuring. These questions were researched beforehand through literature and observations. This is an established method. Still, it is possible that certain feeding practices were not discovered that way, as we concentrated on practices already mentioned in literature and what we have seen. The answers of the participants, in contrast, were then coded inductively without any predefined structure.

We have included this aspect of potential bias in the limitation section.

Reviewer: Discussion in its present form is long and reads like an extension of findings. The sub-headings in the Discussion are helpful and can be possibly used in the results/findings section.

Authors response: Thank you for the advice. We changed some of the headers in the results section to enhance visibility that these are findings and not a discussion

Reviewer: Lastly, culture plays a major role in feeding and childcare practices in countries with collectivist cultures such as Kenya. While this is mentioned in the Discussion section under ‘Perceptions and tradition’, some more engagement with how culture influences feeding practices and its outcomes in the Discussion section might be useful. 

Authors response: Thank you for this recommendation. We reviewed the literature and extended the discussion accordingly.

Best regards

Irmgard Jordan on behalf of all authors.

Reviewer 2 Report

This is a very interesting manuscript on an important topic. It is well written.  I suggest some minor revisions:

1. line 145 - ...encourage the child to eat instead of encourage it to eat          2. line 154 - ...like the larger EatSANE study instead of prevailing study            3. line 170 - ...in instead of on                                                                            4. line 175- ... withdrew instead of withdraw                                                      5. line 224- ... respectively. The practice of ... (Divided a long sentence into 2                         sentences. 

6.  Table 1.   Assistance. "Because he can't share with others and can't feed by herself." Change to "himself".

7. line 236-237- ...need for help in feeding was no not an exclusion criterion to be served on a separate plate.

8. line 242 - Learning social skills were was associated with...

7. Line 246-247.  ...like provide providing company and show showing care

8. Line 252-254. ...practiced also other feeding styles. Solutions to boost a child's appetite includes also includes harmful practices such as offering the child...

9. Table 2.  2. Search for alternatives. Let other people feed. Instead of the mother herself another person took over to feed the feeding

10. Table 2.  2. Boost the appetite - ...offering glucose or multivitamins.  I suggest using the word sugar instead of glucose.

11. Table 2. 4  False promises.  The word "mandazi" is an African donut. 

Table 3. Child is forced to eat; child is left alone to eat. "Interact with child during mealtimes and actively and lovingly encourage it him/her to eat.

12. Page 10, line 313, "hindering to serve". I suggest "hindered serving"

13. page 10, line 326, "...since she is I am the one who serves meals."

14. line 337, "...These include perceived respect for the husband if he was not served first related to serving him first.

15. Page 11, line 353-354, They reported experience and observe experiencing and observing different positive outcomes

16. Page 11, line 370,  fewer children were forced to eat but more assisted to eat more were assisted to eat.

Author Response

Dear Reviewer,

Thank you for your positive response and your valuable input on this manuscript, which has helped us to improve it. We have revised the manuscript considering your and the other reviewer recommendations. We are sorry to have missed so many errors and thank you very much the corrections which we have implemented as suggested. As this study is part of a larger study, we were asked to revise the language in some sections (mainly method) to reduce similarity to a publication we wrote and cited which was developed on a different topic in the same project. We have revised the text and like before refer to the relevant publication without reducing the information needed to understand the study. We hope that the revisions meet with your approval.

Best regards

Irmgard Jordan on behalf of all authors